# An Improved Supervoxel Clustering Algorithm of 3D Point Clouds for the Localization of Industrial Robots

**Zhexin Xie** [1,2]**, Peidong Liang** [2]**, Jin Tao** [1,2]**, Liang Zeng** [1,2]**, Ziyang Zhao** [2]**, Xiang Cheng** [1]**, Jianhuan Zhang** [1]
**and Chentao Zhang** [1,*]

1. Department of Instrumental and Electrical Engineering, Xiamen University, Xiamen 361000, China; zhexinxie@stu.xmu.edu.cn (Z.X.); amy_king_tj@163.com (J.T.); 35120201151533@stu.xmu.edu.cn (L.Z.); chengflying@xmu.edu.cn (X.C.); aeolus@xmu.edu.cn (J.Z.)
2. Fujian (Quanzhou)-HIT Research Institute of Engineering and Technology, Quanzhou 362000, China; lpd0004@hitqz.com (P.L.); zhaoziyang6406@dingtalk.com (Z.Z.)
* Correspondence: zhangct@xmu.edu.cn; Tel.: +86-15959285107

**Abstract:** Supervoxels have a widespread application of instance segmentation on account of the merit of providing a highly approximate representation with fewer data. However, low accuracy, mainly caused by point cloud adhesion in the localization of industrial robots, is a crucial issue. An improved bottom-up clustering method based on supervoxels was proposed for better accuracy. Firstly, point cloud data were preprocessed to eliminate the noise points and background. Then, improved supervoxel over-segmentation with moving least squares (MLS) surface fitting was employed to segment the point clouds of workpieces into supervoxel clusters. Every supervoxel cluster can be refined by MLS surface fitting, which reduces the occurrence that over-segmentation divides the point clouds of two objects into a patch. Additionally, an adaptive merging algorithm based on fusion features and convexity judgment was proposed to accomplish the clustering of the individual workpiece. An experimental platform was set up to verify the proposed method. The experimental results showed that the recognition accuracy and the recognition rate in three different kinds of workpieces were all over 0.980 and 0.935, respectively. Combined with the sample consensus initial alignment (SAC-IA) coarse registration and iterative closest point (ICP) fine registration, the coarse-to-fine strategy was adopted to obtain the location of the segmented workpieces in the experiments. The experimental results demonstrate that the proposed clustering algorithm can accomplish the localization of industrial robots with higher accuracy and lower registration time.

**Keywords:** supervoxel; moving least squares (MLS); instance segmentation

## 1. Introduction

Without the application of industrial robots, it is impossible to accomplish automation and modernization of a manufacturing process in any industrial branch. Due to their strong adaptability and flexibility, industrial robots are commonly assigned for dull, dangerous, and unpleasant jobs to replace human beings, including automatic spray painting [1], welding [2], grinding [3], logistics sorting [4], assembling [5] and so on.

Computer vision, as the eyes of industrial robots, has become a significant component of the robotic system in previous decades. Due to the advantage of obtaining richer physical information about objects, extensive research on 3D vision has been carried out in recent decades. However, 3D segmentation is still one of the most challenging tasks in computer vision. The results of 3D segmentation will directly affect object recognition [6], pose estimation [7] and positioning [5] in the application of industrial robots. The goal of the segmentation process is to group points that belong to the same objects into clusters or sets, where each cluster has similar properties by some criteria. Three-dimensional segmentation methods include the traditional pointwise methods and the segmentation methods based on deep learning, which emerged during those years. The segmentation method based on

deep learning is one of the solutions; however, the training data must be manually labeled, which is more challenging than labeling 2D images [8–10]. It is impractical to adopt the deep learning approach in engineering applications for difficult training data preparation. The pointwise methods are more widely used in engineering projects as no training data are required and they have high adaptability [11]. However, the pointwise methods are inefficient for point clouds with large data volumes. Furthermore, in the application of industrial robots, noises and the adhesion of objects' point clouds are generated for many reasons, including similar and colorless workpieces stacked in a mass, the light reflection, and the shooting angle of 3D cameras. Without eliminating the useless noises and adhesion, it is easy to generate mis-segmentation and under-segmentation.

The supervoxel-based method inspired by superpixels can effectively reduce point clouds' data volume. The superpixel method is widely used in 2D image processing to effectively reduce the computation consumption of the subsequent process [12]. In recent years, the efficient supervoxel method has been introduced to 3D semantic segmentation [13–15]. Supervoxels were applied in a convolution operation (SVConv) by Huang, Ma et al. to effectively accomplish online 3D semantic segmentation [16]. In Sha, Chen et al.'s work, road contours were extracted efficiently and based completely on a supervoxel method without any trajectory data [17]. The Euclidean clustering algorithm was optimized by supervoxels to improve the anti-noise ability of the clustering process by Chen et al. [18]. Li, Liu et al. proposed a multi-resolution supervoxels method to improve accuracy in regions of inconsistent density [19]. Lin, Wang et al. adopted an adaptive resolution for each supervoxel to preserve object boundaries effectively [20]. Although the existing supervoxel segmentation methods have the abilities of data reduction and anti-noise, the adhesion problem is still unsolvable, which results in the low accuracy of segmenting objects, especially in complex industrial applications.

To improve the accuracy and efficiency of 3D instance segmentation under the condition of stacked workpieces with weak texture, a bottom-up clustering method based on supervoxels was proposed. In the supervoxel-based over-segmentation algorithm, moving least squares (MLS) surface fitting was utilized to refine the supervoxel clusters, which can eliminate noises and adhesion. In the merging algorithm, the precise geometric and spatial features are extracted from refined supervoxel clusters, which are generated from over-segmentation. Then, according to the convexity-concavity judgment and the distance metric consisting of feature information, the supervoxel patches are merged to complete 3D instance segmentation.

In summary, the main contributions of this paper are as follows:

1.  An improved supervoxel over-segmentation algorithm with MLS surface fitting was proposed to effectively eliminate the adhesion caused by shooting angles and reflections. Additionally, the over-segmentation method performs data simplification.
2.  A multi-feature metric combined with convexity-concavity judgment was proposed. An adaptive approach was added to this metric to normalize different features. According to the metric, over-segmentation patches can be merged via the proposed merging algorithm.

The organization of this paper is as follows: in Section 2, the proposed methodology is introduced including preprocessing, over-segmentation based on supervoxels and MLS, multi-features region merging. In Section 3, the experimental results with quantitative and visible outputs are demonstrated to analyze the viability and advantages of the proposed method. Finally, the conclusion is summarized in Section 4.

## 2. Methods

The proposed bottom-up method includes data preprocessing, an over-segmentation algorithm, and a region merging algorithm, as shown in Figure 1b. The point clouds are obtained by a binocular structured light camera in Figure 1a. After object instance segmentation in Figure 1b, the objects' point clouds with shape and location information are extracted. As shown in Figure 1c, combined with the sample consensus initial alignment

and iterative closest point (SAC-ICP) registration, bin-picking experiments are conducted
to test the improvements and feasibility of the proposed method.

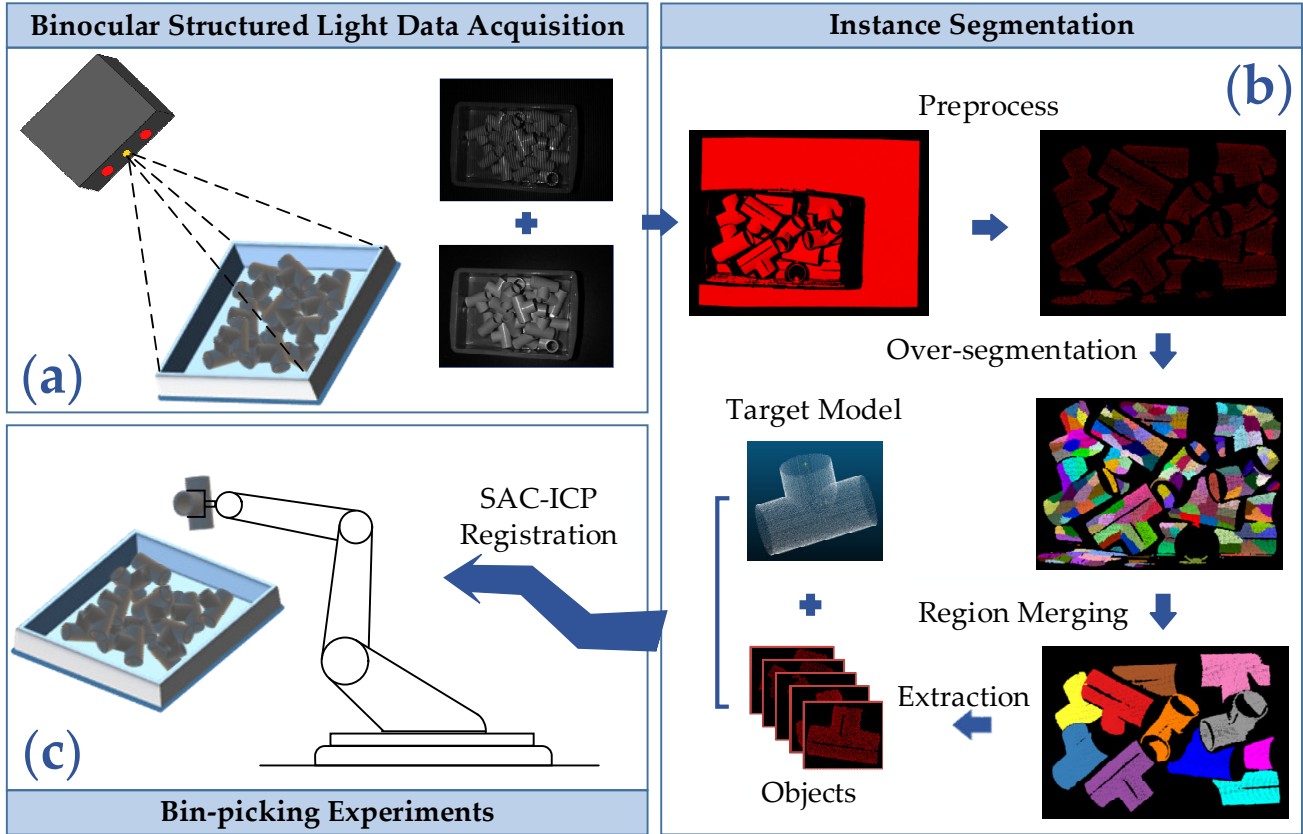

**Figure 1.** The process of our method and its application. Subfigures (**a**–**c**) display the acquisition of
point cloud data, the objects instance segmentation, and bin-picking experiments, respectively.

*2.1. Data Acquisition and Preprocessing*

The workflow of this section is shown in Figure 2, including data acquisition, down
sampling, plane removal, and outlier points removal.

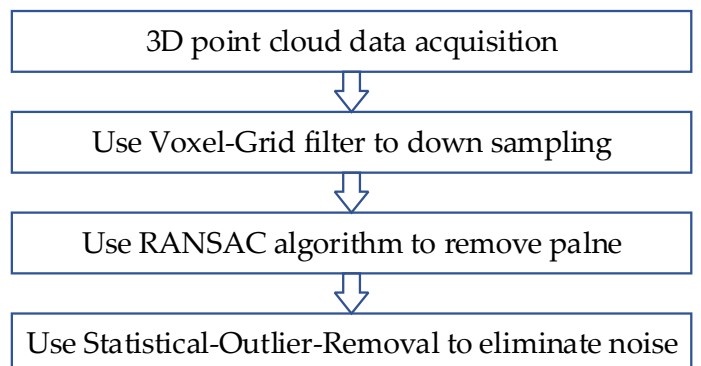

**Figure 2.** The process of data acquisition and preprocessing.

A binocular structured light 3D camera was applied to attain high-quality 3D point
cloud data with a resolution of up to 0.02 mm. The large amount of point cloud data results
in high computational complexity. Thus, the voxel grid algorithm was implemented to
perform down sampling, while maintaining the input data's shape characteristics and geo-
metric properties. Given the voxel grid size, the point clouds can be divided into multiple
voxel grids by octree, as shown in Figure 3. The entire point clouds are approximately

expressed by the centroids of those voxel grids to achieve down sampling. The coordinates (XYZ) of centroids can be calculated as follows:

$$\begin{cases} \overline{x} = \frac{1}{n} \sum_{(x,y,z \in V)} x \\ \overline{y} = \frac{1}{n} \sum_{(x,y,z \in V)} y \\ \overline{z} = \frac{1}{n} \sum_{(x,y,z \in V)} z \end{cases} \tag{1}$$

where $n$ is the total number of points in the voxel $V$.

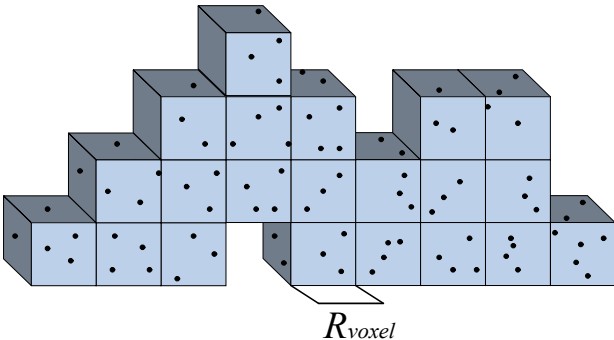

**Figure 3.** Representation of voxels generated by octree. The different cubes represent voxels, the dots represent the original points. The leaf size ($R_{voxel}$) was set at 1 mm in the experiments.

The points include not only objects but also planes and other noisy points that will go against the subsequent instance segmentation. The random sample consensus (RANSAC) algorithm was utilized to remove the plane. The RANSAC algorithm randomly sampled three points as the minimum point set to generate a hypothetical plane in every iteration. Then, the distance between the remaining points and the plane generated by these three points was calculated by the following Formula (2):

$$D = \left| \frac{ax + by + cz + d}{\sqrt{a^2 + b^2 + c^2}} \right| \tag{2}$$

where $a$, $b$, $c$, $d$ are the parameters of the calculated plane equation. For a given distance threshold ($\delta = 2$ mm), the number of points whose distance was below the threshold were counted as inliers. After iteration, the RANSAC algorithm returned to the plane with the highest percentage of inliers.

The statistical outlier removal algorithm was adopted to eliminate the noisy points [21]. After traversing each point's k-nearest ($k = 6$) neighbors, this approach deleted the points whose average distance to their neighbors was more than multiple standard deviations of the mean distance to the query point. The points in accordance with Formula (3) remained.

$$p = \left\{ p \in P \middle| (\mu_p - \sigma_p \cdot std\_mul) \leq \overline{d} \leq (\mu_p + \sigma_p \cdot std\_mul) \right\} \tag{3}$$

where $\overline{d}$ represents the average distance between $p$ and its k-nearest neighbors, $std\_mul$ represents the standard deviation multiple threshold (usually $std\_mul = 1$), $\mu_p$ and $\sigma_p$ are the mean and standard deviation of the Gaussian distribution, which was generated by the average distance between $p$ and the remaining point, respectively.

The results of preprocessing are shown in Figure 4. These steps reduce the number of points in the original data and remove noises. They help to decrease the downstream processing calculation consumption and increase the accuracy of the proposed method. However, there still are noises and adhesion that cannot be eliminated, as shown in Figure 4d, which we aim to remove in the following section.

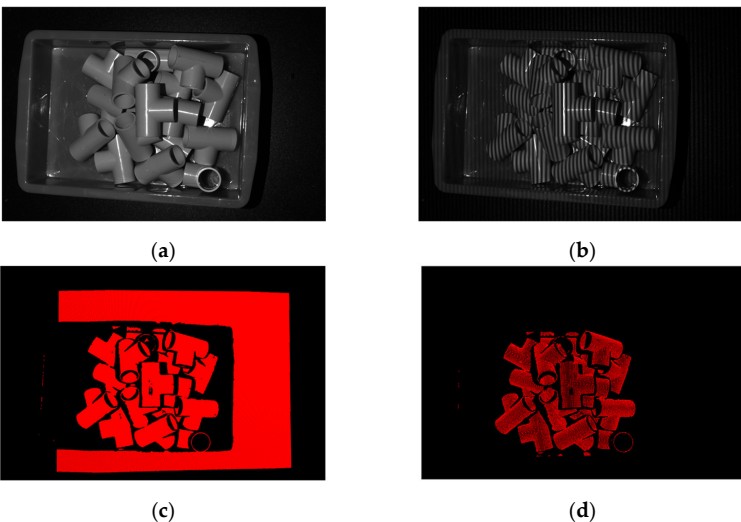

**Figure 4.** The results of preprocessing. (**a**) The original figure without the color of the objects, (**b**) the picture of the objects when structured light was projected, (**c**) the original point cloud we obtained; (**d**) the point cloud after preprocessing.

### 2.2. Over-Segmentation Based on Supervoxels and MLS Surface Fitting

Unsupervised over-segmentation is one of the most widely used point cloud processing methods, which has been extensively used in computer vision. Similar to superpixels, the point cloud is divided into voxel regions with analogous properties by supervoxel segmentation. One of the most widely used supervoxel methods is voxel cloud connectivity segmentation (VCCS) [22]. However, mis-segmentation commonly occurs using VCCS for unclear boundaries. Research has been performed to refine supervoxels. Guarda et al. proposed a C2NO algorithm to generate constant size, compact, nonoverlapping supervoxel clusters [23]. In Xiao et al.'s work, a merge-swap optimization framework was introduced to generate regular, compact supervoxels with adaptive sizes using an energy function [24]. The points that belong to two separate objects are grouped into one cluster in the segmentation of stacked industrial workpieces, which is caused by noises and adhesion. An improved over-segmentation approach was proposed to address this issue based on supervoxels and MLS surface fitting. The noisy points and adhesion, which cannot be removed by preprocessing, can be effectively eliminated. Consequently, the proposed method realizes the goals of minimizing the mis-segmentation occurrence and enhancing the accuracy of workpiece instance segmentation. The process of this method is shown in Figure 5. The details of the proposed method will be elaborated on in this section.

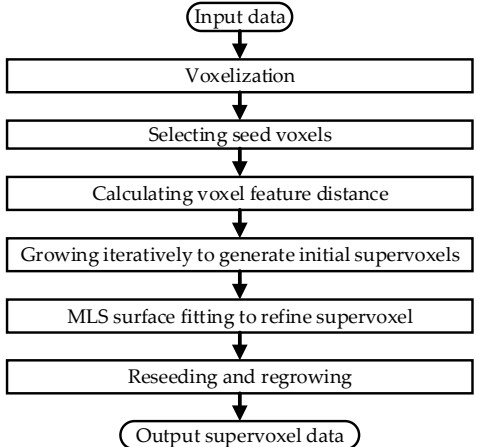

**Figure 5.** The process of an improved over-segmentation approach based on supervoxels and MLS surface fitting.

### 2.2.1. The Selection of Seed Voxels

For the given resolution of a voxel, the over-segmentation algorithm begins with the voxelization that is generated from the point cloud by octree. The process of supervoxel over-segmentation is similar to polycrystalline nuclear crystallization of the supersaturated saline solution, where all the crystal nuclei grow simultaneously. Therefore, seed voxels need to be selected to initialize the supervoxels after the voxelization. The spatial relationship among those voxels was created by building an adjacency graph on 26-adjacent of the voxel. Assuming that each seed is evenly distributed in the three-dimensional space, the voxels most approximated to the centers of the given seed resolution are selected as the candidates of seed voxels.

Some candidates of seed voxels isolated from their neighbors need to be deleted. The seed voxels where there is not a sufficient number over *min_n* of voxels surrounding them in the search area should be removed. The filter criterion is as follows:

$$R_{search} = \frac{1}{2} R_{seed} \tag{4}$$

$$min\_n = \frac{\pi R_{search}^2}{20 R_{voxel}^2} \tag{5}$$

$$sd = \{sd \in SD | n > min\_n\} \tag{6}$$

where $R_{search}$ represents the search radius of the seed voxels, $R_{seed}$ represents the revolution of seed voxels, which decides the distance between adjacent supervoxels and $R_{voxel}$ represents the size of voxels generated by voxelization. $R_{seed}$ should be much larger than $R_{voxel}$; otherwise, the seeds will not be selected correctly, which may cause mis-segmentation. *sd* represents the seed voxels; only the candidates that fit Formula (6) will remain as the initial supervoxels. In Equations (4)–(6), only the parameters including $R_{seed}$ and $R_{voxel}$ need to be assigned based on the size of objects. Other parameters are able to be calculated according to these two parameters. Figure 6 shows the geometric representation of those parameters.

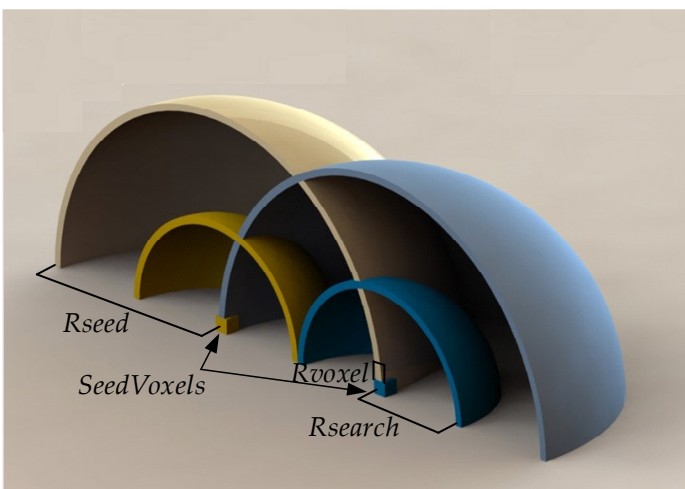

**Figure 6.** Parameters of seed selection.

### 2.2.2. Voxel Feature Distance

In our implementation of supervoxel generation, the voxel feature distance is utilized to determine the similarity of seed voxels and their adjacent voxels. First, the spatial distance is normalized to limit the search scope of every clustering iteration. The algorithm will stop searching when it approaches the cluster center of the adjacent supervoxel by using a maximum range of $\sqrt{3} R_{seed}$ to normalize the spatial distance. Then, the normal difference is calculated, which characterizes the degree of surface bending. Thus, the boundary properties of 3D voxel data can be represented by spatial distance and normal.

$$D_s = \sqrt{(x_i - x_j)^2 + (y_i - y_j)^2 + (z_i - z_j)^2} \tag{7}$$

$$D_n = 1 - \left| \vec{N_i} \cdot \vec{N_j} \right| \tag{8}$$

$$D = \sqrt{\frac{\lambda D_s{}^2}{3R_{seed}{}^2} + \mu D_n{}^2} \tag{9}$$

where $x$, $y$, $z$ are spatial coordinates, $\vec{N}$ is the normal of a voxel, $D_s$ and $D_n$ are the spatial distance and normal difference, respectively. $D$ is the fusion distance of two features, where $\lambda$ and $\mu$ are the parameters that allocate the influential proportion of two feature distances.

By iteratively traversing the adjacent voxels of all the initial supervoxels, the voxels will melt into their neighboring supervoxels, according to spatial distance and normal distance. The lower voxels are searched and processed layer by layer until all the adjacent voxels of supervoxels are traversed. Additionally, after updating the cluster centers of the supervoxels, supervoxels regrow until the cluster centers are stable or this algorithm reaches the maximum iterations.

### 2.2.3. MLS Surface Fitting

Due to the noises and adhesion of point clouds, surface fitting is adopted to refine the supervoxels. The least squares method has a widespread application in curve and surface fitting. It has been improved by many researchers, including total least squares (TLS), recursive least squares (RLS), weighted least squares (WLS), generalized least squares (GLS), partial least squares (PLS) and segmented least squares (SLS). However, in the cases of the large amount and irregular, scattered distribution of point cloud data, the above-mentioned methods are not suitable on account of their global approximation strategies. The moving least squares (MLS) [25] method is utilized, which is a local approximation to represent the surface of supervoxel clusters. Compared with the traditional least square method, every point in the fitting region will be projected to the locally weighted fitting surface in the MLS method. On a local subdomain of the fitting region, the fitting function is defined by the following equation:

$$f(x,y) = \sum_{i=1}^{6} \alpha_i(x) p_i(x,y) = p^T(x,y)\alpha(x) \tag{10}$$

$$p(x,y) = \left[ 1, x, y, x^2, xy, y^2 \right]^T \tag{11}$$

$$J = \sum_{I=1}^{n} w(x - x_I)[f(x,y) - z_i]^2 \tag{12}$$

where $n$ represents the number of points in the local reference domain of a given radius at the target point. $w(x - x_I)$ is a weighted function, which guarantees the increasing contribution to optimization function $J$ with decreasing distance from the sampling point to the target point.

To consider MLS's sensitivity to outliers, radius outlier removal is adopted to eliminate the isolated points while avoiding excessive fitting deviation before performing MLS fitting. After adding the MLS fitting filter, the adhesion of separate objects' point clouds caused by the structured-light projection angle and stacking can be removed without affecting the shape characterization of the objects. The two examples of the denoising results are shown in Figure 7. There are significant changes that occurred in the boxes, where the adhesion of two separate objects' point clouds can be removed, while the point clouds still convey shapes.

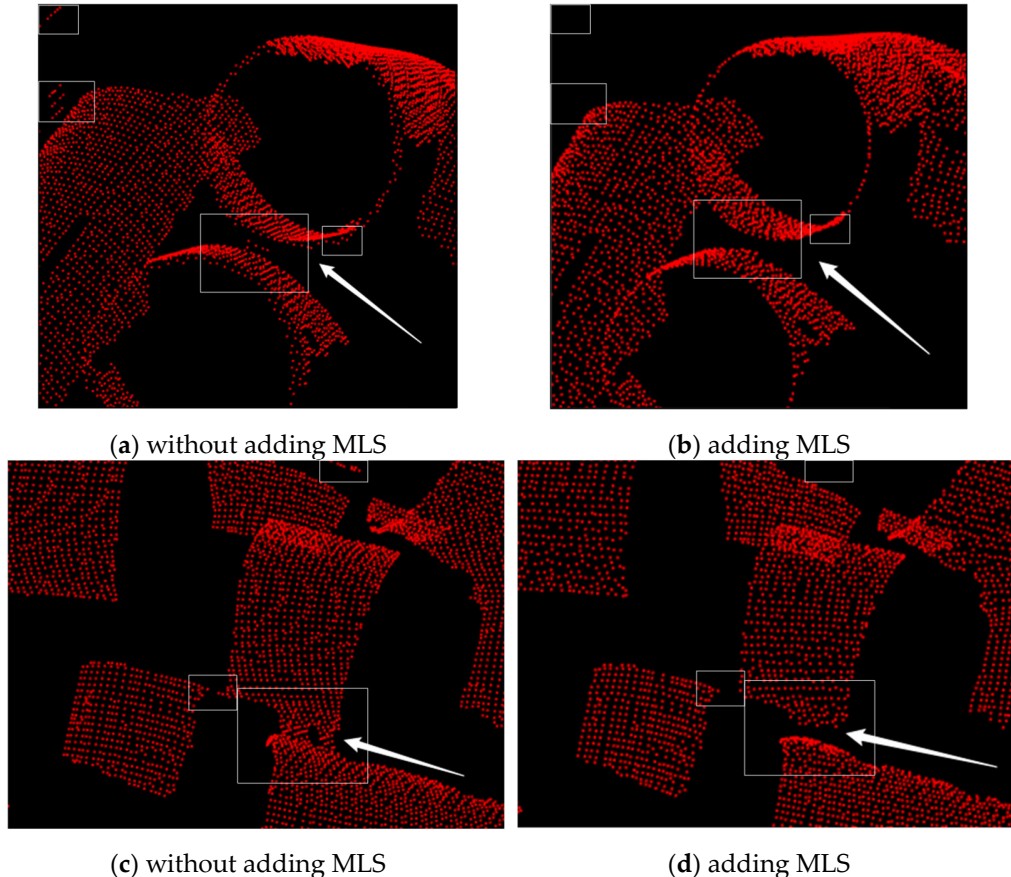

(**a**) without adding MLS             (**b**) adding MLS

(**c**) without adding MLS             (**d**) adding MLS

**Figure 7.** The results of the MLS surface fitting filter. (**a**,**b**) are the point clouds before and after adding the MLS surface fitting filter in the same scene, respectively. (**c**,**d**) are the point clouds in the other scene.

### 2.3. Region Merging Based on Multi-Feature with Convexity Judgment

The patches produced by over-segmentation should be merged into object clusters. The supervoxel patches contain precise geometric and other information about the objects. If we give additional constraints based on the geometric characters and structure relationships between patches, the instance segmentation will be accomplished without a training dataset.

The distance metric was proposed to decide whether the patches are clustered or not, which plays a significant role in the merging algorithm. The distance metric is a fusion of the following two features: a geometric feature distance $\delta_G$ that represents the geometric distance between any two adjoining supervoxel patches; a spatial distance $\delta_D$ that captures the Euclidean distance of any two adjacent patches' centroids.

As shown in Figure 8a, for any two patches $p_s$ and $p_t$, their centroids are represented by $x_t$ and $x_s$, their normal vectors by $\vec{n_s}$ and $\vec{n_t}$ and the unit vector laying on the line connecting the two centroids by $\vec{C_{st}}$ ($\vec{C_{st}} = \frac{x_t - x_s}{\|x_t - x_s\|}$). To represent the locational and geometrical relationship of the two patches, the feature distance $\delta_D$ and $\delta_G$ can be described as follows:

$$\delta_D(p_s, p_t) = \| x_t - x_s \| \tag{13}$$

$$\delta_G(p_s, p_t) = \frac{\| \vec{n_s} \times \vec{n_t} \| + | \vec{n_s} \cdot \vec{C_{st}} | + | \vec{n_t} \cdot \vec{C_{st}} |}{3} \tag{14}$$

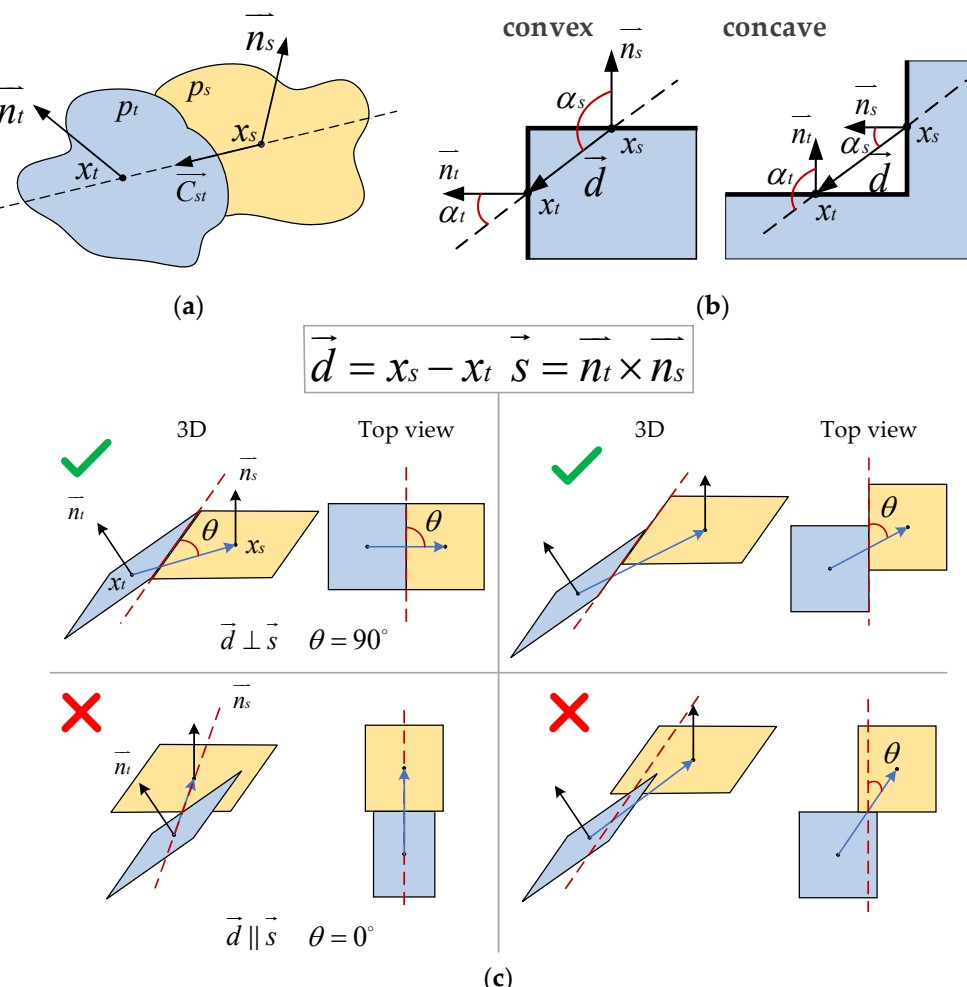

**Figure 8.** Geometrical relationship of two patches. (**a**) Features of two patches. (**b**) Geometrical meaning of convex-concave criterion. (**c**) Geometrical meaning of the sanity criterion.

The locally convex connected patches (LCCP) [26,27] method is a segmentation method based on the concave and convex relations between two patches, which was also exploited in other works [28,29]. In our work, the geometrical feature distance $\delta_G$ is incorporated with a convexity criterion inspired by the LCCP method. The new $\delta_G$ is defined by the following equation:

$$\delta_G(p_s, p_t) = \begin{cases} 0.5\delta_G, & if(\alpha_s - \alpha_t < 0 \cup \angle\left(\vec{n_s}, \vec{n_t}\right) < 1°) \cap \theta \geq intersect\_threshold \\ \delta_G, & otherwise \end{cases} \quad (15)$$

$$intersect\_threshold = 60°\left\{1 + e^{-0.25°\left[\angle(\vec{n_s}, \vec{n_t}) - 60°\right]}\right\}^{-1} \quad (16)$$

As illustrated in Figure 8b,c, if the angle $\alpha_s$, $\alpha_t$ between the normal vectors $\vec{n_s}$, $\vec{n_t}$ of the two adjoining patches and the line connecting their centroids $\vec{d}$ is deemed as convex, the geometric distance $\delta_G$ is halved. The two patches with a valid convex property are more probable to be merged into parts of the same object. Two patches are evaluated as convex if and only if they comply with both the convex-concave criterion and the sanity criterion. If $\alpha_s < \alpha_t$ or $\angle\left(\vec{n_s}, \vec{n_t}\right) < 1°$ i.e., two patches are almost parallel; it is regarded that they are convex in the convex-concave criterion. However, the convex property must be validated by the sanity criterion to be valid. Two surfaces are disconnected when there is only a singular connection between them. Only if the angle between the cross product of normal

vectors $\vec{s}$ and the line connecting centroids $\vec{d}$ is large enough, i.e., $\theta \geq intersect\_threshold$, the two patches with connectivity can then be confirmed as convex.

　　Owing to the geometric data and spatial distance data being intrinsically different types of data, the two features need to be transformed into a unified domain for normalization. Therefore, the proposed distance value between two supervoxel patches $p_s$ and $p_t$ can be described as follows:

$$\delta(p_s, p_t) = T_G(\delta_G(p_s, p_t)) + T_D(\delta_D(p_s, p_t)) \tag{17}$$

where $T_G$ and $T_D$ are two transformations defined to normalize two feature distances into unified ranges between 0 and 1. Any of the two feature items in Formula (17) can be changed; consequently, an adaptive value $\lambda$ as weight is proposed to feed the specific needs of different applications. Presumptively, $\delta_G$ and $\delta_D$ have unknown distributions with unknown means $\mu_G$ and $\mu_D$. We can then define the adaptive $\lambda$ and the transformations $T_G$, $T_D$ by the following equation:

$$\lambda = \frac{\mu_G}{\mu_D + \mu_G} \tag{18}$$

$$T_G(g) = (1 - \lambda)g \tag{19}$$

$$T_D(d) = \lambda d \tag{20}$$

The process of the region merging algorithm is shown in Figure 9.

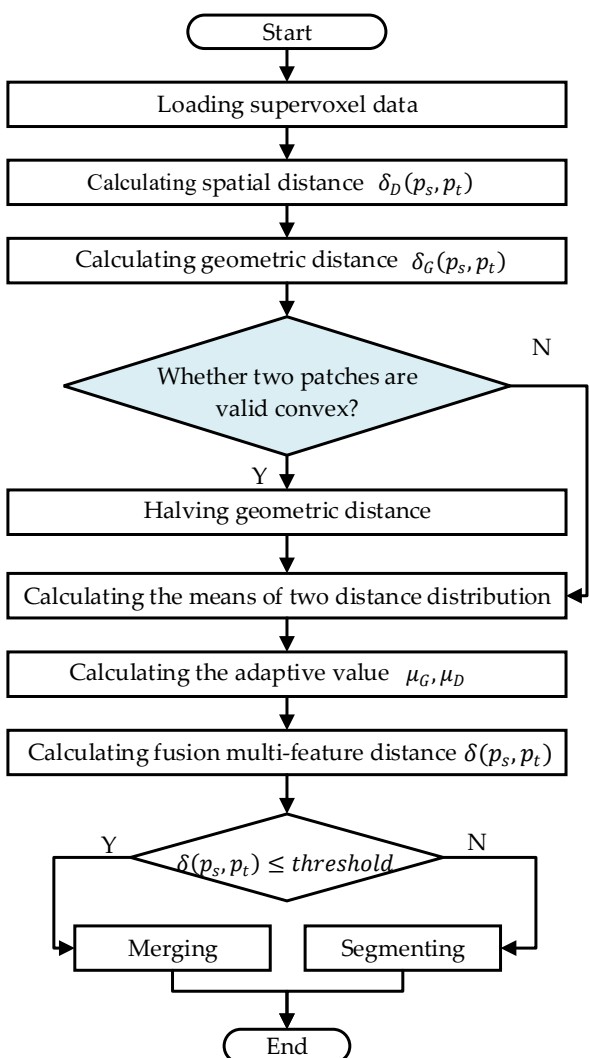

**Figure 9.** The process of the region merging algorithm.

*2.4. Evaluation*

To evaluate the segmentation results, some experimental performance indicators are described as follows.

The ground-truth partition $G = \{G_1, G_2, G_3, \ldots, G_n\}$ is defined as a set of artificial labeled points set $G_i$, and the segmentation result $S = \{S_1, S_2, S_3, \ldots, S_m\}$ is defined as a set of regions generated by the algorithm in the same point cloud. Additionally, $N_G = n$ represents the number of ground-truth regions. The precision is defined to evaluate the segmentation result of our algorithm compared with the ground-truth, as described by the Formula (22), which is as follows:

$$TP_i = G_i \cap S_i \tag{21}$$

$$Pre_i = \frac{TP_i}{TP_i + FP_i} = \frac{TP_i}{S_i} \tag{22}$$

where $TP_i$ represents the number of points for an object in region *i* and accurately segmented as the object in the segmentation result. $TP_i$ is calculated by figuring the overlap point cloud between $G_i$ and $S_i$. $FP_i$ represents the number of points actually for an object but not segmented as the object in region *i*.

Due to the noises and other reasons, the artificial annotated objects may not be quite correct. So, the workpiece with a precession of larger than 95% will be regarded as successfully segmented. The number of successfully segmented workpieces is defined as $N_T$. The recognition rate is defined by Formula (23):

$$Reg = \frac{N_T}{N_G} \tag{23}$$

The results of instance segmentation will significantly and directly affect positioning accuracy, related to registration. Consequently, to test the validity and efficiency of the proposed algorithm in industrial robot applications, registration experiments were performed. Many point cloud registration algorithms have been proposed including singular value decomposition (SVD) [30], random sample consensus (RANSAC) [31], normal distributions transform (NDT) [32], sample consensus initial alignment (SAC-IA) [33], iterative closest point (ICP) [34] and its improved algorithm [35,36]. SAC-IA coarse registration and ICP fine registration algorithm were adopted for their precision and high efficiency. Firstly, the SAC-IA algorithm was utilized to perform coarse registration, using the fast point feature histogram description (FPFH) [33] as the point cloud feature description. The transformation matrix obtained by SAC-IA was used as the initial matrix in the ICP algorithm. Then, the target point cloud was aligned to the template point cloud by minimizing the distance iteratively to attain the fine matrix. The fitness score, i.e., the mean square error (MSE) between the target workpiece and the template workpiece, was calculated using Formula (24), which is as follows:

$$MSE = \frac{1}{m} \sum_{i=1}^{m} (\hat{p}_i - q_i)^2 \tag{24}$$

where $\hat{p} = \{\hat{p}_i | i = 1, 2, 3 \ldots\}$ and $Q = \{q_i | i = 1, 2, 3 \ldots\}$ represent the points in the target point clouds after translation and the points in the template point clouds, respectively. *m* is the number of point pairs. The objects with a fitness score below 1.2 mm$^2$ are defined as high matching objects. Therefore, the high registration rate means the proportion of high matching workpieces.

## 3. Experimental Results and Discussion

The experimental setup is shown in Figure 10. The experiments were conducted in the following two parts: instance segmentation compared with ground-truth; segmentation performance tests combined with SAC-ICP registration in industrial application. The experiment platform was Intel Core i7-8750, with 8G memory, Windows 10 64-bit operating

system, VS2015VC++win64 console application, and open source point library PCL 1.9.1. Three kinds of workpieces were taken for the experiments to test the feasibility in different scenes, as shown in Figure 11.

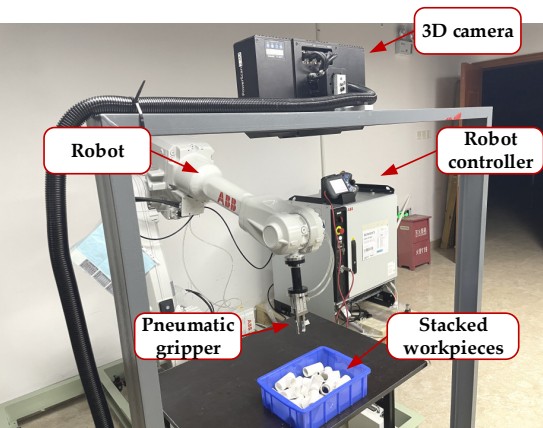

**Figure 10.** The experimental setup. A binocular structured light 3D camera is used to obtain high-quality point clouds of stacked workpieces. The ABB IRB 2600-20 robot with pneumatic grippers at the end is shown in the figure. I/O programming can control the clamping state of the pneumatic gripper to grasp workpieces. The robot is controlled by the controller.

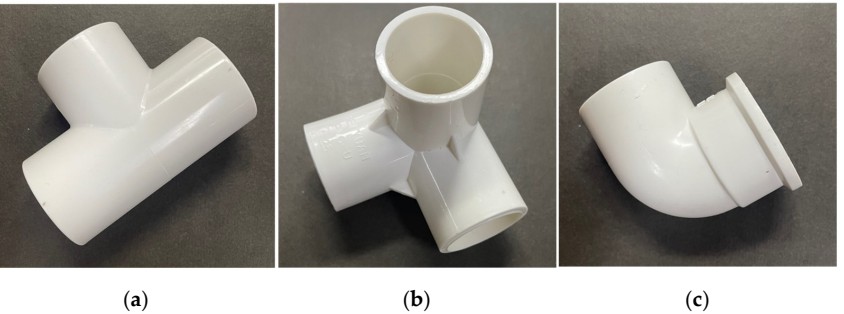

|  (**a**)  |  (**b**)  |  (**c**)  |

**Figure 11.** Three kinds of workpieces for experiments. Subfigures (**a**–**c**) are Tee pipe 1, Tee pipe 2, and Two-way elbow, respectively.

### 3.1. Instance Segmentation Experiments

Ten experiments were conducted to demonstrate the data simplification ability of our over-segmentation method, whereas VCCS in the same revolution cannot simplify point cloud data. Table 1 illustrates that the simplifying radio in ten experiments all reach over 65%. The reason why our method can simplify data is that the over-segmentation based on MLS can reduce the useless and noisy points.

**Table 1.** The simplifying example results of our method ($R_{voxel} = 1 \, \text{mm}$ , $R_{seed} = 20 \, \text{mm}$ ).

| Experiments | Processed Data Size | VCCS | | Proposed Method | |
|---|---|---|---|---|---|
| | | After [1] | Simplifying Radio | After [1] | Simplifying Radio |
| 1 | 30,629 | 30,629 | 0 | 21,936 | 71.618% |
| 2 | 28,528 | 28,528 | 0 | 20,300 | 71.158% |
| 3 | 34,051 | 34,051 | 0 | 26,228 | 77.026% |
| 4 | 31,330 | 31,330 | 0 | 21,110 | 67.380% |
| 5 | 25,327 | 25,327 | 0 | 18,074 | 71.363% |
| 6 | 25,758 | 25,758 | 0 | 18,028 | 69.990% |
| 7 | 31,360 | 31,360 | 0 | 21,714 | 69.241% |
| 8 | 36,012 | 36,012 | 0 | 25,342 | 70.371% |
| 9 | 41,715 | 41,715 | 0 | 33,315 | 79.863% |
| 10 | 33,146 | 33,146 | 0 | 23,797 | 71.794% |

[1] The data size after over-segmentation.

Ten experiments were conducted in three groups of different workpieces to study the performance of the proposed method compared with other methods. The parameters we adopted are listed in Table 2. Different thresholds were set for different kinds of workpieces with the best accuracy. However, the same voxel size was utilized in different methods, including search radius.

**Table 2.** The parameters of methods.

| Parameters | Euclidean | VCCS + LCCP | Proposed Method |
|---|---|---|---|
| Voxel size (search radius) | 2 mm | 2 mm | 2 mm |
| Seed size | \ | 8 mm | 8 mm |
| Threshold [1] (highest accuracy) | \ | 10°/5°/20° | 0.25/0.25/0.35 |

[1] Tee pipe 1/Tee pipe 2/Two-way elbow.

The segmentation results of different methods are shown in Figure 12. The proposed method can segment stacked workpieces accurately, while under-segmentation occurs in other methods. The results of the segmentation accuracy and recognition rate are listed in Table 3. The exchanged tests were performed to analyze different contributions of our bottom-up method, including VCCS combined with our merging method and our over-segmentation method combined with LCCP. The results demonstrate that the average precision of the proposed method reached up to 0.988, 0.984, 0.988 in Tee pipe 1, Tee pipe 2 and Two-way elbow, respectively. The recognition rates were 0.936, 0.975, 0.958, respectively. The results illustrate that the proposed method is more accurate than other methods. The proposed over-segmentation method plays a significant role in enhancing the segmentation accurate rate.

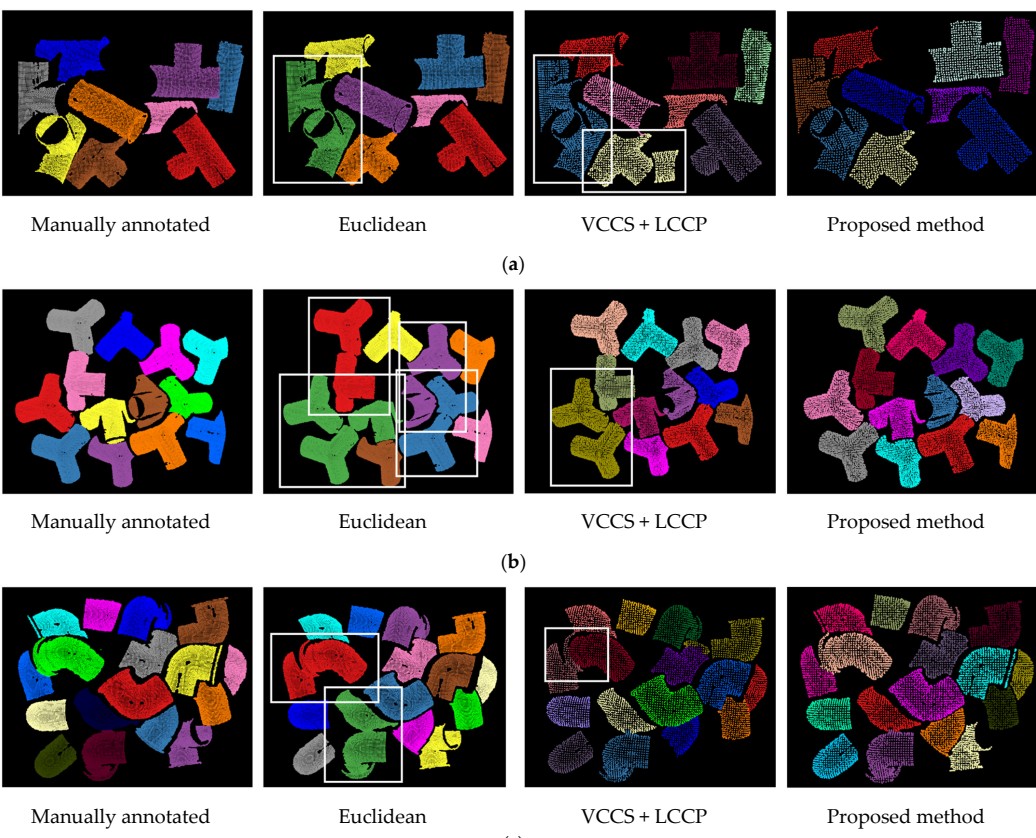

**Figure 12.** Results of the comparison of different workpieces. (**a**) Tee pipe 1. (**b**) Tee pipe 2. (**c**) Two-way elbow. Mis-segmentations are demonstrated by white boxes. (Points were enlarged for visualization).

**Table 3.** The comparison of performance.

| Workpieces | Methods | Pre-Average | Reg-Average |
|---|---|---|---|
| Tee pipe 1 | Euclidean | 0.835 | 0.705 |
| | VCCS + LCCP | 0.943 | 0.831 |
| | VCCS + our merging | 0.915 | 0.780 |
| | Our over-segmentation + LCCP | 0.934 | 0.868 |
| | Proposed method | 0.988 | 0.936 |
| Tee pipe 2 | Euclidean | 0.796 | 0.643 |
| | VCCS + LCCP | 0.899 | 0.828 |
| | VCCS + our merging | 0.906 | 0.836 |
| | Our over-segmentation + LCCP | 0.917 | 0.855 |
| | Proposed method | 0.984 | 0.975 |
| Two-way elbow | Euclidean | 0.908 | 0.813 |
| | VCCS + LCCP | 0.928 | 0.845 |
| | VCCS + our merging | 0.926 | 0.840 |
| | Our over-segmentation + LCCP | 0.974 | 0.942 |
| | Proposed method | 0.988 | 0.958 |

To check the position accuracy after segmentation, an analysis of the mean errors between the segmented workpieces' centroids and the artificial annotated point cloud in the XYZ-axis was performed. The comparisons of the mean errors using different methods in the same point cloud data are shown in Figure 13. In each group of Figure 13a–c, the mean errors of all the workpieces in the XYZ-axis are calculated for every experiment. The mean errors of the workpieces' centroids are volatile in other methods (blue and black lines). The mean error in the Euclidean method even reaches higher than 19 mm in the X-axis of Tee pipe 2. The mean errors of the segmented workpieces' centroids are mostly below 2 mm in the proposed method, which is lower and more stable than the other methods. Consequently, the proposed method can maintain the shape characteristics and locational information of workpieces; simultaneously, the point cloud data can also be simplified.

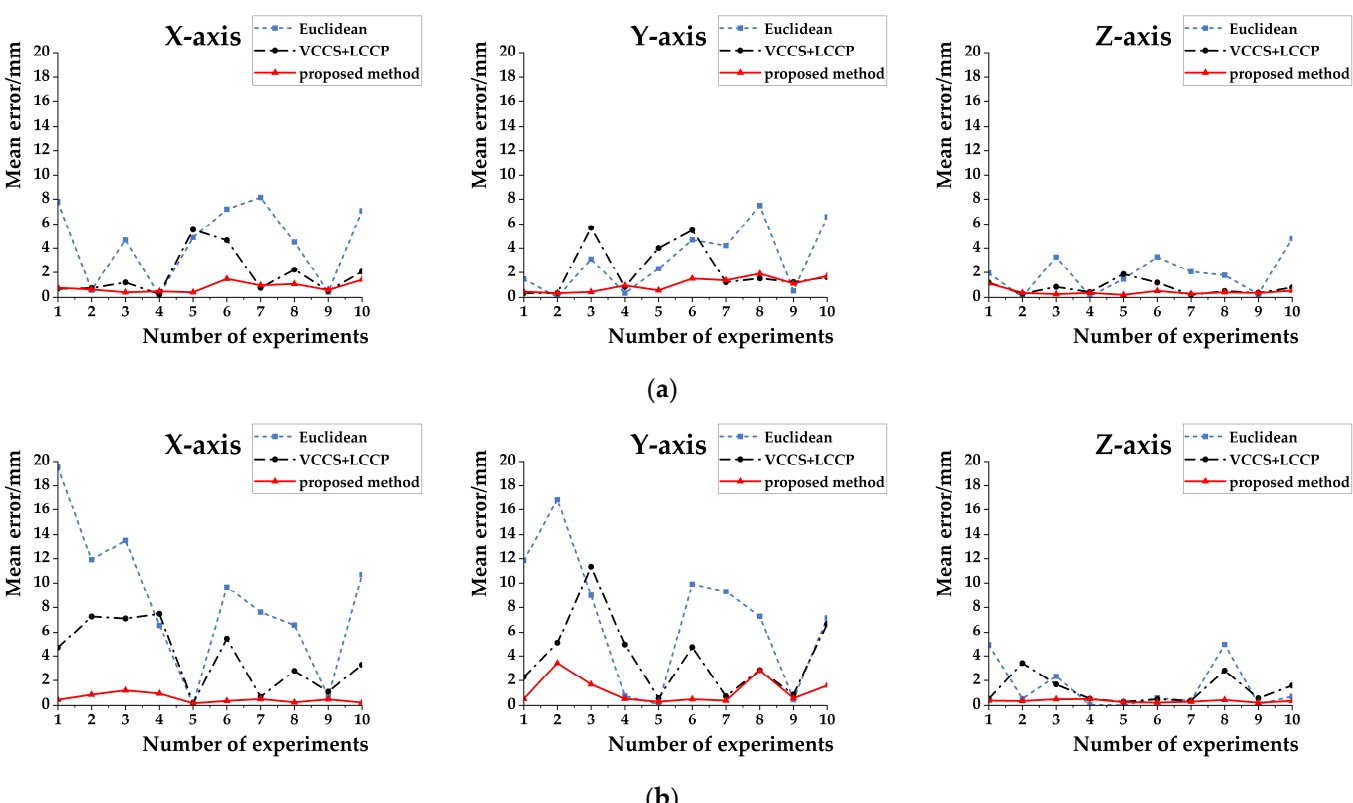

**Figure 13.** *Cont.*

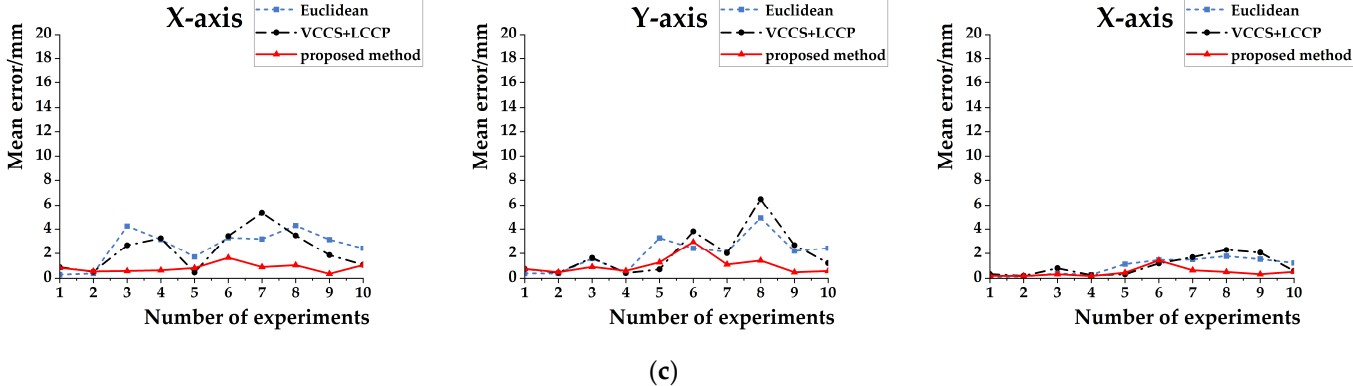

(**c**)

**Figure 13.** The mean error comparison. (**a**) Tee pipe 1. (**b**) Tee pipe 2 (**c**) Two-way elbow. Each group of (**a**–**c**) includes three figures of the mean errors on the XYZ-axis.

### 3.2. SAC-ICP Registration Experiments

The registration results are shown in Figure 14. The registration results compared with other methods in the same registration parameters are listed in Table 4. The fitness scores of different workpieces are diverse because of the object shape and registration algorithm. Compared with other methods, the target point clouds and the template point clouds were matched more accurately in the proposed method for accurate instance segmentation. Owing to the effective simplification of data, the proposed method can save more registration consuming time with low error and high registration rates.

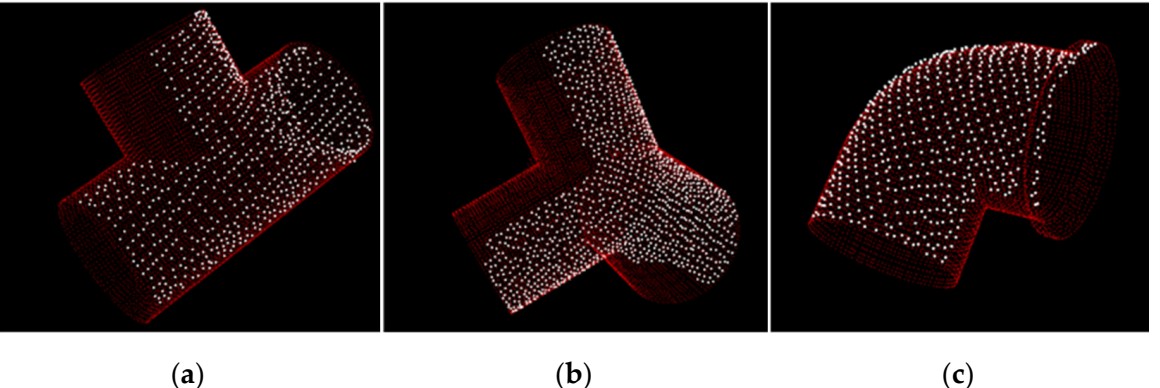

(**a**)　　　　　　　　　(**b**)　　　　　　　　　(**c**)

**Figure 14.** The registration results. (**a**) Tee pipe 1. (**b**) Tee pipe 2 (**c**) Two-way elbow. The red points in the figures are the template point clouds of different workpieces. The white points are the segmented point clouds of workpieces.

**Table 4.** The registration results.

| Workpieces | Methods | MSE | High Registration Rate | Running Time/ms |
|---|---|---|---|---|
| | Euclidean | 36.546 | 0.423 | 10,564.733 |
| | VCCS + LCCP | 4.632 | 0.670 | 2535.233 |
| Tee pipe 1 | VCCS + our merging | 8.055 | 0.568 | 2171.403 |
| | Our over-segmentation + LCCP | 4.402 | 0.699 | 1996.968 |
| | Proposed method | 2.003 | 0.749 | 1892.472 |
| | Euclidean | 101.968 | 0.299 | 13,506.750 |
| | VCCS + LCCP | 24.064 | 0.649 | 3399.876 |
| Tee pipe 2 | VCCS + our merging | 19.626 | 0.639 | 2944.655 |
| | Our over-segmentation + LCCP | 12.095 | 0.743 | 2719.517 |
| | Proposed method | 1.595 | 0.862 | 2368.400 |

**Table 4.** *Cont.*

| Workpieces | Methods | MSE | High Registration Rate | Running Time/ms |
|---|---|---|---|---|
| | Euclidean | 5.590 | 0.572 | 12,193.280 |
| | VCCS + LCCP | 5.471 | 0.578 | 1965.480 |
| Two-way elbow | VCCS + our merging | 5.712 | 0.546 | 2371.985 |
| | Our over-segmentation + LCCP | 2.641 | 0.620 | 1746.778 |
| | Proposed method | 2.559 | 0.708 | 1595.524 |

## 4. Conclusions

In this paper, an improved instance segmentation method based on supervoxels for the localization of industrial robots has been proposed, which can process point cloud data more accurately, robustly, and effectively. An over-segmentation algorithm with MLS surface fitting was presented, which generates supervoxel patches, while eliminating noisy points and point clouds' adhesion by refinement. Additionally, the adaptive region merging algorithm based on multi-features and convex-concave judgment was performed to accomplish instance segmentation. The experimental results demonstrate the feasibility and stability of the proposed method for application in industrial robots. Compared with other traditional methods, the proposed method achieves the instance segmentation of workpieces with higher precision and recognition rate under the complex condition of multiple similar stacked objects. Furthermore, the registration time can be reduced due to the data simplification of the proposed method. In future work, the energy function will be considered based on the proposed method to avoid boundary overlap. Additionally, the supervoxel-based over-segmentation clustering will be further developed for the application in semantic segmentation.

**Author Contributions:** Conceptualization, Z.X. and P.L.; methodology, Z.X.; software, Z.X., Z.Z. and J.T.; validation, L.Z., X.C., J.Z. and C.Z.; formal analysis, Z.X., P.L., J.T. and C.Z.; investigation, Z.X.; resources, Z.Z. and C.Z.; data curation, L.Z. and C.Z.; writing—original draft preparation, C.Z., Z.X. and P.L.; writing—review and editing, C.Z. and J.Z.; visualization, Z.X.; supervision, X.C., J.Z. and C.Z. All authors have read and agreed to the published version of the manuscript.

**Funding:** This research was funded by the National Key Research and Development Program of China under Grant No. 2018YFB1305703, the University-Industry Collaboration Project of Fujian Province under Grant No. 2019H6003, and the Fundamental Research Funds for the Central Universities under Grant No. 20720210063.

**Data Availability Statement:** The data presented in this study are available on request from the corresponding author. The data are not publicly available due to privacy.

**Conflicts of Interest:** The authors declare no conflict of interest.

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
