# Peer review of "An Improved Supervoxel Clustering Algorithm of 3D Point Clouds for the Localization of Industrial Robots"

_electronics, doi:10.3390/electronics11101612_

Round 1

Reviewer 1 Report

The paper is well organized and the results are clear, the only drawback is filtered which is introduced in equations 4 to 6. The question is how the parameters of the filter are assigned for different objects?

Author Response

Dear reviewer:

We would like to appreciate you for the detailed review of our manuscript. The insightful comments and suggestions are helpful to improve our manuscript. Please find our resubmission of the manuscript with revisions included. We have made revisions and explanations one by one. For the details and the responses from the authors, please see the Word attachment.

Thank you very much.

Sincerely Yours,

Reviewer 2 Report

The paper is very well. However, the following comments need to be addressed to further improve the quality of the paper.

  1. There are several abbreviations which are not defined in the paper. E.g. VCCS, LCCP
  2. The choice of moving least squares among the several least techniques available in literature is not properly justified.
  3. The Pseudo code or flowchart of the methodology will help in great understanding of authors.
  4. Why only 10 trails or experiments?
  5. Add future part of the work
  6. Check all grammar and spelling mistakes 

Author Response

(The authors gave the same response as above.)

Reviewer 3 Report

Please state the novelty of the research clearly.

Please follow the journal's format for labelling subfigures (a) , (b) and (c) in Figure 1.

"where" after equations should be with small w instead of capital W.

'T" should be used at the beginning of line 134 instead of "t".

Please correct the formatting for line 168.

Please explain what is D in equation (9).

Please consider changing line 276 to "3. Experimental Results and Discussions"

Please do not start a sentence with "And" such as in line 210 (and other places too).

All abbreviations need to be spelled out in full term before being used. E.g. RANSAC in line 125.

Please consider placing all the formula under method instead of putting them under the results.

Figure 11 needs to be further elaborated and the effectiveness of the proposed method needs to be emphasized in this elaboration.

Please consider revising Table 2 so that it is more readable. Suggestion : the left column to be the parameters, and the the top row to be the methods.

Please change "Lots" in line 350 and other places in the paper to "Many"

Some of the years for the papers on in the reference were not bold.

The paper needs to be sent for proofreading before sending the final version.

Author Response

(The authors gave the same response as above.)
